# Differences in hospital admissions practices following self-harm and their influence on population-level comparisons of self-harm rates in South London: an observational study

C Polling  ,[1,2] Ioannis Bakolis,[3] Matthew Hotopf,[1,2] Stephani L. Hatch[1]

[1]Psychological Medicine, Institute of Psychiatry, Psychology and Neuroscience, King's College London, London, UK
[2]South London and Maudsley NHS Foundation Trust, London, United Kingdom
[3]Department of Biostatistics and Health Informatics, Institute of Psychiatry, Psychology & Neuroscience, King's College London, London, UK

**Correspondence to**
Dr C Polling;
catherine.polling@kcl.ac.uk

## ABSTRACT

**Objectives** To compare the proportions of emergency department (ED) attendances following self-harm that result in admission between hospitals, examine whether differences are explained by severity of harm and examine the impact on spatial variation in self-harm rates of using ED attendance data versus admissions data.

**Setting** A dataset of ED attendances and admissions with self-harm to four hospitals in South East London, 2009–2016 was created using linked electronic patient record data and administrative Hospital Episode Statistics.

**Design** Proportions admitted following ED attendance and length of stay were compared. Variation and spatial patterning of age and sex standardised, spatially smoothed, self-harm rates by small area using attendance and admission data were compared and the association with distance travelled to hospital tested.

**Results** There were 20 750 ED attendances with self-harm, 7614 (37%) resulted in admission. Proportion admitted varied substantially between hospitals with a risk ratio of 2.45 (95% CI 2.30 to 2.61) comparing most and least likely to admit. This was not altered by adjustment for patient demographics, deprivation and type of self-harm. Hospitals which admitted more had a higher proportion of admissions lasting less than 24 hours (54% of all admissions at highest admitting hospital vs 35% at lowest). A previously demonstrated pattern of lower rates of self-harm admission closer to the city centre was reduced when ED attendance rates were used to represent self-harm. This was not altered when distance travelled to hospital was adjusted for.

**Conclusions** Hospitals vary substantially in likelihood of admission after ED presentation with self-harm and this is likely due to the differences in hospital practices rather than in the patient population or severity of self-harm seen. Public health policy that directs resources based on self-harm admissions data could exacerbate existing health inequalities in inner-city areas where these data may underestimate rates relative to other areas.

## BACKGROUND

Self-harm, through both self-injury and overdose, affects over 6% of the population in England at some point in their lifetime[1]

### Strengths and limitations of this study

► This study links clinical records data to administrative Hospital Episode Statistics enabling the creation of a dataset of emergency department (ED) attendances which better represents service use for self-harm than admissions data alone and is not available from routine data.

► Use of individual-level data allows investigation of the roles patient factors, type of self-harm and length of stay play in the differences between hospitals.

► The focus on a specific local context has enabled us to use clinical record data and incorporate the knowledge of clinicians working within the services investigated to understand the findings but means that estimates of between hospital differences are probably conservative.

► ED attendance data show a less biassed picture of the spatial patterning of self-harm than admissions, however, it may still be influenced by the differences in likelihood of presentation to ED between populations.

and results in an estimated 220 000 emergency department (ED) presentations[2] and over 100 000 hospital admissions[3] each year in England. Increasing public health and government interest is being focused on self-harm both as a concern in itself and as an important risk factor for future suicide.[4] Substantial geographical variations occur in the rates of self-harm[5,6] and suicide[7] across the country and within local authority areas. National policy in England emphasises the need for the development of suicide and self-harm prevention plans by local authorities using 'localised real time data'[8] to determine need and appropriate targeting of interventions. However, attempt to understand variations in self-harm rates in this way is hampered by the limited data routinely available.

In most countries, the only nationwide, routinely available data on service use for self-harm is based on hospital admissions. Only the Ireland has established nationwide routine monitoring of presentations to EDs with self-harm that do not result in admission.[9] In England, research data on ED use following self-harm are routinely collected in hospitals in Manchester, Derby and Oxford[10] but outside of these centres, the only reliably coded routine data are Hospital Episode Statistics (HES) for rates of admissions to hospital.[11] This is an administrative dataset reported by all National Health Service (NHS) hospitals in which all hospital stays have International Classification of Diseases, version 10 (ICD-10) diagnostic codes attributed to them, including codes related to self-harm.[12] Hence, much research considering how and why rates of self-harm vary between areas has relied on admission data alone. Admissions data also feature prominently in national and local public health planning. In England, it is included as an indicator in the Public Health Outcomes Framework (PHOF)[11] as well as Public Health England's area profiles[13] whose explicit purpose is to allow monitoring and comparison of areas.

Underlying the use of admissions data for work comparing areas is the assumption that, while it contains only a minority of all self-harm that occurs, individuals from different demographic groups and geographical areas who self-harm in a similar way are equally likely to feature it in. The usual presumption is that these data represent the most severe cases of self-harm,[5] implying a uniformity of severity in the cases admitted to different hospitals. There is a reason to question these assumptions. Work using data from Manchester, Oxford and Derby has compared the rates of presentation to general hospitals with self-harm using ED attendance data to the rates using HES admissions data. It found that the ratio of admission rate to attendance rate varied between the three centres.[14] If such differences are due to variations in admission practices between hospitals, they have the potential to introduce substantial biases. Using admissions data could produce misleading estimates of relative rates for comparisons between areas and distort our understanding of area-based health inequalities. A better understanding of biases within data on admissions for self-harm could inform their use within public mental health in England and highlight areas for investigation in similar datasets internationally.

This study uses data from four hospitals in South East London as a case study to investigate whether there are differences between hospitals in the likelihood of admission for individuals attending EDs following self-harm. London is of particular interest for work on self-harm as admissions data suggest that it has much lower rates of self-harm than the national average[15] despite having high levels of deprivation and social fragmentation, area-level factors that have consistently been shown to be associated with high area rates of self-harm.[16] Previous work has found this counterintuitive pattern replicated within London, with areas closer to the city centre having

lower rates of self-harm admission, a finding that remains unexplained.[6]

Unlike previous studies, we have linked a research dataset of ED attendances to HES admissions data at the individual level. This allows the study to investigate how rates of admission vary between different hospitals and to account for differences due to the sociodemographics of the population served or type of self-harm seen. It also considers the impact of variations in the severity of self-harm presenting to each hospital by examining average length of admission and the distance individuals have to travel to get to the ED, as work in Ireland has suggested presentation after minor self-injury is more common when individuals live closer to the hospital.[9] The study goes on to compare the conclusions reached about the spatial patterning of self-harm when ED attendance rather than admission data were used.

## METHODS

### Aims

This study aims to compare the proportions of ED attendances for self-harm that result in admissions that would be included in HES self-harm admissions data for four hospitals in South East London between 2009 and 2016 and average length of stay for admitted patients, adjusting for the age, sex, ethnicity and economic deprivation of those presenting and the type of self-harm. It then aims to compare the amount of variation between areas and geographical patterning of self-harm rates across the study area when (1) ED attendances and (2) admissions are used as the definition of self-harm and to examine whether any differences persist after ED attendance rates are adjusted for the distance individuals have travelled to get to the ED.

### Setting

The study area, shown in figure 1, consists of four London boroughs: Lambeth, Southwark, Lewisham and Croydon, which stretch from central London in the north to the edge of Greater London in the south and have a combined population of 1.2 million. It is served by four hospitals with EDs, King's College Hospital (KCH), St Thomas' Hospital (STH), University Hospital Lewisham (UHL) and Croydon University Hospital (CUH), each of which is run by a separate NHS hospital trust. Secondary mental healthcare for the whole area, including liaison psychiatry services in all four EDs, is provided by South London and Maudsley NHS Foundation Trust (SLaM). When examining geographical patterning, rates were calculated for Lower Super Output Areas (LSOA, average population 1700), of which there are 728 in the study area. These are lowest level geography available in HES data and were used because there is great heterogeneity in area type across short distances within London.

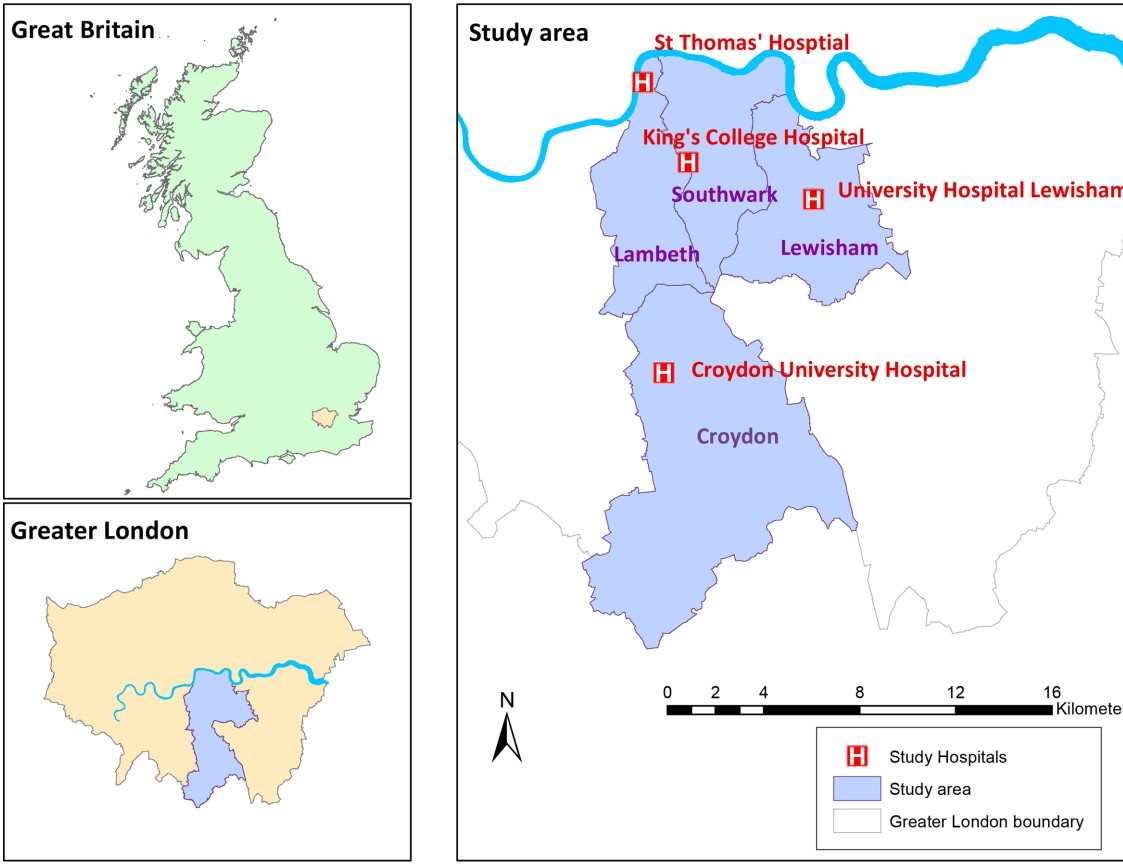

Boundaries: Office of National Statistics, 2001 Census: Digitised Boundary Data (England and Wales) [English Lower Layer Super Output Areas, 2001] UK Data Service Census support. Downloaded from: https://borders.ukdataservice.ac.uk.

**Figure 1** Study area.

## Data

Data were accessed via the Clinical Records Interactive Search system (CRIS), a case register created from the anonymised electronic patient record of SLaM,[17] which is linked to HES admissions data. CRIS contains HES data for all individuals who have ever had contact with SLaM services plus all individuals living in the four boroughs of the study area, regardless of whether they have ever had contact with mental health services.

## Outcome

### ED attendance

ED attendances by individuals aged 11 or older recorded as living within the study area at the time of attendance at an ED were identified by combining CRIS and HES data. The full methods have been published elsewhere.[18] In brief, HES Accident and Emergency data were used to identify attendances to the EDs of the four study hospitals. Additional attendances were identified from CRIS records of referrals to ED based liaison teams. CRIS was then used to identify any free-text entries in the SLaM electronic record made by a mental health liaison team or recorded as made in the ED between the date and time of ED arrival and 12 hours after the date and time of ED departure. Entries containing keywords relating to self-harm and suicidality (see online supplementary material 1) were extracted and manually coded according to

whether the ED attendance was for an act of self-harm and the type of self-harm. Throughout the study period, all four EDs had 24 hours liaison teams, a policy of referring all self-harm presentations to them and of recording all referrals and their reason in the electronic record even if patients did not wait to be seen.[18]

In a small proportion of cases, individuals attending EDs for self-harm may be admitted but not have any psychiatric assessment within the time window specified, usually because they were too physically unwell for assessment. To ensure these cases were not missed, linked HES Admitted Patient Care data were checked and any admissions via ED given ICD-10 diagnostic codes X60-X84 (self-harm) that did not already feature in the dataset were added.

Part of the dataset for 2009–2013 has previously been validated against an audit dataset created in EDs through a combination of forms competed by psychiatric liaison teams and searches of the ED patient record.[18] The dataset created from electronic health records performed similarly to the audit dataset, detecting 77% of all attendances and 82% of all individual patients, with no differences found in the age, sex, ethnicity or marital status of those detected versus those missed.

### HES record of admission

Admission for self-harm was extracted from the linked HES APC dataset. We used the definition of admission

for self-harm used by Public Health England for its 'Emergency Hospital Admissions for Intentional Self-Harm' indicator[19]: an emergency (unplanned) admission with a 'CAUSE' ICD-10 code in the range X60–X84 (intentional self-harm). These admissions were matched to the corresponding attendances where they related to the same individual at the same hospital with an admission date on or 1 day after the date of ED attendance. This resulted in 44 admissions being matched to more than one attendance, where an individual had attended 2 days in a row and been admitted from the second attendance. In these cases, the first attendance was corrected to show it had not resulted in admission.

### Population denominator

Population denominator data were taken from Office of National Statistics (ONS) midyear estimates.[20] Due to differences in the reporting geographies between HES and ONS data, six LSOAs had to be merged into three to make data comparable, hence 725 areas were used in analyses.

The study had access to data for all four hospitals that principally serve the study area. However, in areas at the edge of the study area individuals may also attend EDs at hospitals in neighbouring boroughs. HES ED data for everyone living in the study area were used to determine the proportion of attendances to EDs for any reason (excluding stand-alone minor injuries and walk-in centres) that were to study EDs. This proportion was used to weight the denominator population of each LSOA when calculating rates, so that rates were not misleadingly low at the edges of the study area.

### Confounders

Individual age, sex, ethnicity and LSOA of residence were taken from CRIS, supplemented by HES data where CRIS was incomplete. Area-level deprivation was measured using the Index of Multiple Deprivation 2010,[21] a composite measure summarising multiple dimensions of deprivation at LSOA level in England. Distance to ED was measured from LSOA centroid to closest ED using ArcGIS software.

### Statistical analysis

#### Individual-level analysis of admission following attendance

All attendances were included in the analyses, with admission as the outcome. A Poisson regression model with robust error variance was used as the high prevalence of admission in the dataset made ORs produced from a logistic regression difficult to interpret meaningfully[22] This model was two level to account for clustering at the individual level where there were repeat attendances within the dataset. Risk ratios (RRs) for admission for each hospital were calculated and adjusted for potential confounders identified a priori: age (in 5-year bands), sex, ethnicity, type of self-harm and deprivation. Wald tests were used for significance. Length of admission was grouped into <24 hours, 1, 2, 3, 4 or 5 or more days and

treated as ordinal in analyses as its distribution was very skewed. Differences between hospitals were tested with a Kruskal-Wallis test. These analyses were carried out in STATA V.15.1.

### LSOA-level analyses of rates of self-harm

For calculation of LSOA rates of self-harm, individuals' first admission and first attendance in the dataset were included. This was because a small number of individuals in the dataset have a large number of admissions and attendances which would have had a disproportionate influence on the rates for the small area in which they reside. Age and sex-standardised rate ratios (SRRs) for self-harm admission and ED attendance alone and adjusted for distance to hospital were calculated for each LSOA. SRRs were smoothed using a Besag-York-Mollie Bayesian disease mapping model,[23] which includes separate spatially structured and unstructured area-level random effects, to account for overdispersion and spatial structure. Smoothing reduces the influence of random noise given the low counts in individual areas and adjusts estimates for spatial autocorrelation.

For each model, the amount of variation in LSOA SRRs was quantified using the 90% quantile ratio (QR90). This is the ratio of the SRR for the area at the 95th centile to the SRR for the area at the 5th centile and so describes the scale of variation in residual relative risk between the top and bottom 5% of areas. The residual SRRs after spatial smoothing and adjustment, which represent the remaining variation, were mapped to display spatial patterning and plotted to allow comparison of residual variation between analyses.

Analyses were carried out in R V.3.2.2. Bayesian models were run using Markov Monte Carlo Chain simulation in OpenBUGS V.3.2.3 using the R2OpenBUGs routine. Results were mapped in ArcMap V.10.6.

### Patient and public involvement

This study forms part of a wider project examining rates of self-harm in London. This project has been discussed with the National Institute for Health Research Maudsley Biomedical Research Centre (BRC) Service User Advisory Group who provided advice on the overall aims of the project. No patients were involved in the planning or design of this study. The findings have been discussed with local community groups and public health teams and will be further disseminated through the BRC's patient and public involvement activity.

## RESULTS

During the study period, 20 750 attendances to study EDs made by 12 577 individuals living in the study area were identified. Of these, 7614 attendances (37%) by 4801 individuals resulted in an admission that was coded as self-harm in HES admission data. The majority of individuals (9557, 76%) attended only once during the study period while the small group of people with more than

five attendances (368, 3%) accounted more than a fifth of all attendances (4393, 21%).

## Individual-level analysis of admission following attendance

There were 20 750 ED attendances with self-harm, 7614 (37%) of which resulted in admission. Table 1 shows distribution of age, sex, ethnicity, types of self-harm and deprivation for ED attendances and admissions. Table 2 shows the proportions of ED attendances that resulted in admissions by hospital and the distribution of lengths of stay. There were substantial differences in the proportions being admitted between the four hospitals in the study area: compared with KCH, which admitted the lowest proportion, the RRs for the other three hospitals were all two or above, with the greatest difference for CUH (RR 2.45, 95% CI 2.30 to 2.61, p<0.0001). The effect sizes were almost unchanged when adjusted for the sex, age and ethnicity of the individuals attending, the type of self-harm they presented with and the level of deprivation in the local areas in which they lived, for example, the RR for CUH versus KCH reduced to 2.41 (2.27 to 2.55, p<0.0001). The distribution of lengths of admissions also varied significantly between hospitals (Kruskal Wallis test for equality, p=0.0001). Notably more than half (53.6%) of the admissions to CUH were for less than 24 hours, compared with 34.5% at KCH. Conversely, 28.3% of KCH admissions lasted 2–4 or 5+ days compared with 15.5% of CUH admissions. The other two hospitals, STH and UHL, lie between these two extremes.

## LSOA-level analyses of rates of self-harm

Modelling smoothed, age and sex standardised rates of self-harm for LSOAs based on attendance and admission data separately demonstrated that rates based on admissions data had greater spatial variation. The QR90 for SRRs based on attendance was 2.87 (95% credible interval (CrI), 2.65 to 3.13) while that for admissions was 4.51 (3.99 to 5.12). Plots of the residual SRRs, included in online supplementary figure 1, demonstrate that LSOAs with both high and low rates estimated using admissions data tend to have less extreme rates when attendance data is used.

Figure 2 visualises the effect of these differences in the attendance and admissions datasets when the geographical patterning of self-harm is considered. LSOA rates of self-harm admission are clustered with, in general, lower rates of self-harm in areas closer to the city centre. As is shown in online supplementary figure 2 and previous work,[6] these patterns are not explained by area deprivation but in fact strengthen when deprivation is adjusted for. When ED attendance data are used, there is less difference between rates of self-harm in inner and outer-city areas. Some inner-city areas with apparently below average rates of self-harm using admission data are shown to have above average rates of ED attendance for self-harm, although an overall pattern of lower rates in the inner-city remains, all be it with smaller differences in standardised rates. This is shown when the SRRs for

the quintile of LSOAs closest to the city centre versus the furthest in the two datasets are compared. For admissions, the SRR is 0.65 (0.40 to 1.02) (while the CrI for

| Table 1 Characteristics of attendances at EDs and admissions following self-harm at four general Hospitals in South East London 2009–2016 | | |
|---|---|---|
| | ED attendances (%) | Admissions in HES (%) |
| Total | 20 750 | 7614 |
| Sex by age | Missing=9 | Missing=2 |
| Males | | |
| Total | **7686 (37.0)** | **2695 (35.4)** |
| 11–15 | 198 (2.6) | 81 (3.0) |
| 16–19 | 526 (6.8) | 140 (5.2) |
| 20–24 | 1111 (14.5) | 366 (13.6) |
| 25–34 | 1875 (24.4) | 651 (24.2) |
| 35–64 | 3778 (49.2) | 1359 (50.4) |
| 65+ | 198 (2.6) | 98 (3.6) |
| Females | | |
| Total | **13 055 (62.9)** | **4917 (64.6)** |
| 11–15 | 1147 (8.8) | 645 (13.1) |
| 16–19 | 2180 (16.7) | 754 (15.3) |
| 20–24 | 2303 (17.6) | 746 (15.2) |
| 25–34 | 2788 (21.4) | 993 (20.2) |
| 35–64 | 4347 (33.3) | 1629 (33.1) |
| 65+ | 290 (2.2) | 150 (3.1) |
| Ethnicity | missing=293 | missing=104 |
| White | 14 277 (68.8) | 5353 (70.3) |
| Mixed | 795 (3.8) | 315 (4.1) |
| Asian | 883 (4.3) | 355 (4.7) |
| Black | 3248 (15.7) | 1081 (14.2) |
| Other | 1254 (6.0) | 406 (5.3) |
| Type of self-harm | | |
| Overdose | 14 512 (69.9) | 6492 (85.3) |
| Self-injury | 4841 (23.3) | 742 (9.7) |
| Overdose and self-injury | 697 (3.4) | 197 (2.6) |
| Other | 700 (3.4) | 183 (2.4) |
| Index of multiple deprivation | | |
| Least deprived quintile | 2314 (11.2) | 922 (12.1) |
| 2 | 3440 (16.6) | 1340 (17.6) |
| 3 | 4641 (22.4) | 1665 (21.9) |
| 4 | 4885 (23.5) | 1747 (22.9) |
| Most deprived quintile | 5470 (26.4) | 1940 (25.5) |

ED, emergency department; HES, Hospital Episode Statistics.

**Table 2** Length of admission and RRs for admission following attendance at EDs for self-harm in South East London, 2009–2016, by hospital attended

| | Attendances to ED (%) | Admission in HES (%) | Proportion admitted | Mean stay (days) | Length of admission* (%) | | | | Unadjusted RR admission† (95% CI) | Adjusted‡ RR admission† (95% CI) |
|---|---|---|---|---|---|---|---|---|---|---|
| | | | | | <24 hours | 24–48 hours | 2–4 days | 5+ days | | |
| Total | 20 750 | 7614 | 0.37 | 1.21 | 3544 (46.6) | 2557 (33.6) | 1130 (14.8) | 383 (5.0) | | |
| Hospital | | | | | | | | | | |
| KCH | 7106 (34.2) | 1407 (18.5) | 0.20 | 1.90 | 486 (34.5) | 522 (37.1) | 296 (21.0) | 103 (7.3) | 1.00 | 1.00 |
| UHL | 4446 (21.4) | 2033 (26.7) | 0.46 | 1.24 | 903 (44.4) | 697 (34.3) | 318 (15.6) | 115 (5.7) | 2.31 (2.16 to 2.47) | 2.25 (2.12 to 2.40) |
| CUH | 5947 (28.7) | 2886 (37.9) | 0.49 | 0.92 | 1546 (53.6) | 894 (31.0) | 342 (11.9) | 104 (3.6) | 2.45 (2.30 to 2.61) | 2.41 (2.27 to 2.55) |
| STH | 3251 (15.7) | 1288 (16.9) | 0.40 | 1.09 | 609 (47.3) | 444 (34.5) | 174 (13.5) | 61 (4.7) | 2.00 (1.86 to 2.15) | 2.02 (1.89 to 2.17) |

*Kruskal-Wallis test for equality of populations, p=0.0001.
†Wald test, p<0.0001.
‡Adjusted for age, sex, ethnicity, type of self-harm and index of multiple deprivation of residence.
CUH, Croydon University Hospital; ED, emergency department; HES, Hospital Episode Statistics; KCH, King's College Hospital; RR, risk ratio; STH, St Thomas' Hospital; UHL, University Hospital Lewisham.

this estimate crosses 1, the effect size is very similar to that found in previous work on the same geographical area which used data from more years and hospitals (0.67 (0.48 to 0.89)),[6] suggesting that the wide CrI is due to imprecision from a smaller sample size rather than indicating an absence of effect. In this analysis, a smaller dataset was used so that it matched the ED attendance data available while for attendances it is 0.84 (0.60 to 1.13) suggesting admissions data overestimate the effect of proximity to the city centre in lowering rates. Adjusting for the distance individuals had to travel to reach their closet ED makes little difference to the spatial patterning seen when ED attendance data are used. This is reflected in the SRR for self-harm ED attendance for each 1 km increase in LSOA distance from hospital of 0.96 (95% CrI 0.91 to 1.01).

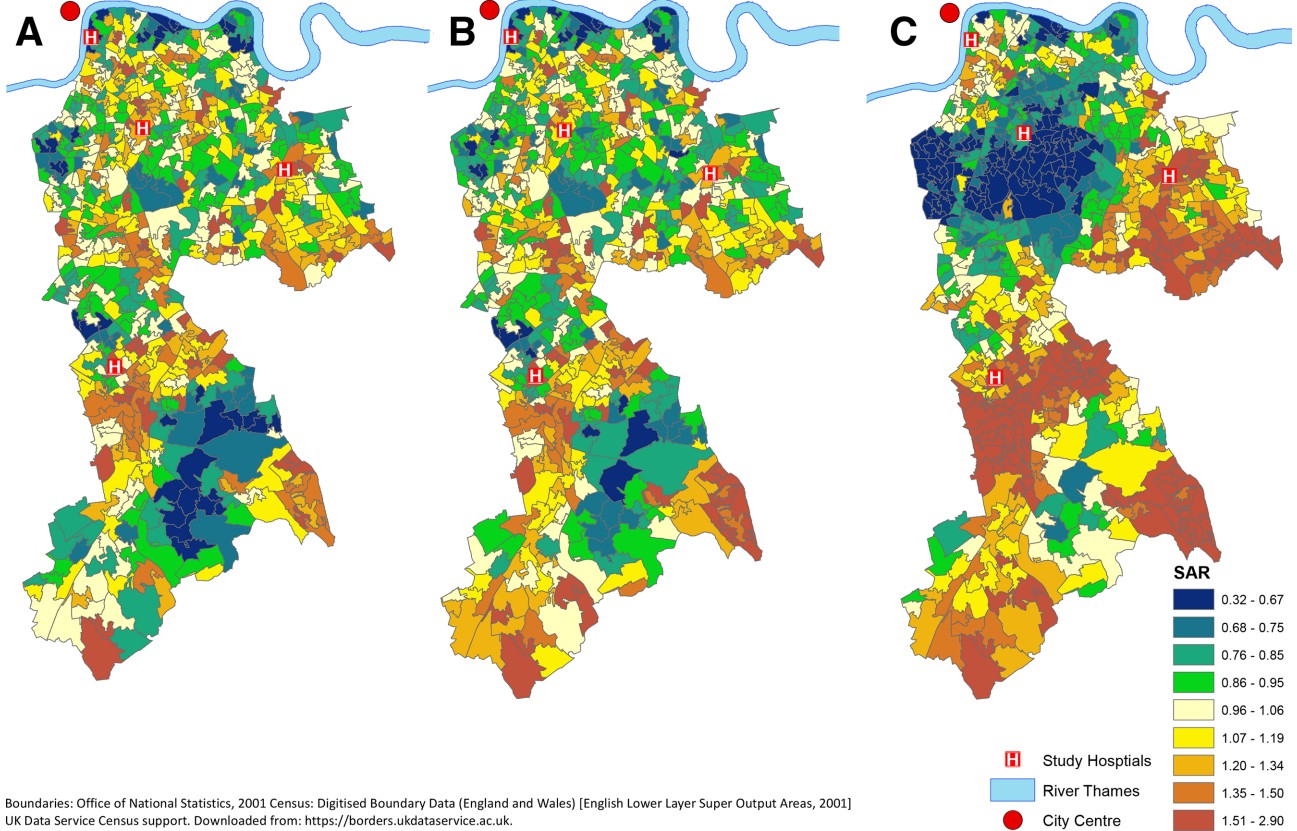

Boundaries: Office of National Statistics, 2001 Census: Digitised Boundary Data (England and Wales) [English Lower Layer Super Output Areas, 2001]
UK Data Service Census support. Downloaded from: https://borders.ukdataservice.ac.uk.

SAR
- 0.32 - 0.67
- 0.68 - 0.75
- 0.76 - 0.85
- 0.86 - 0.95
- 0.96 - 1.06
- 1.07 - 1.19
- 1.20 - 1.34
- 1.35 - 1.50
- 1.51 - 2.90

H Study Hospitals
River Thames
City Centre

**Figure 2** (A) First attendances (B) first attendances adjusted for distance to hospital and (C) first admissions by individuals aged 11+ for self-harm 2009–2016 by lower super output area, standardised for age and sex.

## DISCUSSION

### Principal findings

In this South East London case study area, the HES admission data widely used to represent self-harm rates contain only around one-third of all hospital treated self-harm. Importantly, the likelihood that someone attending an ED with self-harm will be admitted and feature in the HES admission dataset varies substantially according to which ED they attend, with one study hospital almost two and a half times more likely to admit than another. This echoes previous findings of differences in the ratio of attendance rates to admission rates between other English cities[14] and extends them by demonstrating that it is not explained by differences in the demographics of those attending, the deprivation of the areas the EDs served or the type of self-harm people were presenting with. This strengthens the case that differences reflect a difference in practices between the hospitals.

Comparison of the lengths of stay between hospitals suggests that the difference also is not explained by differences in the severity of the self-harm presenting. The hospital with the lowest proportion admitted had the longest average length of stay and the lowest proportion of very short (under 24 hours) admissions, while the hospital with the highest proportion admitted had more than half of its admissions lasting less than 24 hours. This points to differences between hospitals in how likely they are to admit less severe cases.

There were substantial differences in the spatial patterning of self-harm rates seen when different data sources were used. The pattern of clustering of low rates of self-harm in inner-city areas and higher rates of self-harm in areas further from the city centre seen when admission data are used becomes less marked when ED attendance data are used instead. In particular, there are many inner-city areas that appear to have low standardised rates of self-harm using admissions data that are shown to have average or even high rates using ED attendance data. The absence of an association between LSOA self-harm ED attendance rates and distance travelled to hospital, as well as the length of stay findings described above, suggest these differences are not due to the shorter travel times to EDs for individuals in inner-city areas encouraging use of ED services for more minor self-harm that might not use hospital services at all in other areas.

Overall, this study demonstrates that hospitals vary substantially in the likelihood that someone attending ED with self-harm will be admitted, and this is probably not dictated by the severity of self-harm. Discussions with staff within the psychiatric liaison services from the four hospitals studied during the study period (S Cross 2019, personal communication, 2 April: G Ranjith 2019, personal communication, 3 March) have highlighted largely policy-based potential explanations for the difference in admissions seen. For example, hospitals vary in the established ways of managing patients awaiting psychiatric assessment in the ED who were likely to breach national ED waiting time targets[24] and the accepted locations for patients to receive brief courses of treatment. There may also be more general differences, for example, hospitals facing greater general demand on beds may have greater severity thresholds for admission.

### Strengths and limitations

The lack of reliable routine data on ED attendances for self-harm means that this study could only be done in a context where ED attendance could be ascertained in a different way. The availability of clinical records linked to HES data at the individual level within CRIS has allowed us to investigate the potential role patient factors, type of self-harm and length of stay play in differences between hospitals. A focus on a specific local context has also enabled us to use the knowledge of clinicians working within the services investigated to understand the findings.

At a national scale, there are likely to be greater differences in practices between the different hospitals than those seen within one area, so this case study is probably a conservative estimate of the effects of such biases. The specific findings related to a particular geographical area, South East London, however, the limitations it highlights in the use of admissions data to monitor and understand self-harm are relevant across the national area covered by the HES data and in other international contexts where routine admissions data are used.

The use of mental health clinical records to ascertain ED attendances for self-harm will miss some cases. Previous validation work with part of this dataset suggests it detected a similar proportion of ED attendances to an audit dataset based on staff in the ED filling out forms and manual checking of ED notes for cases of self-harm and there were no differences in those detected or missed based on patient demographics.[18] Nonetheless, the implication is that the true number of attendances is higher, so the proportions admitted are likely to be an overestimate. It is possible that despite liaison services in the four EDs being provided by the same mental health trust with apparently uniform referral policies, a greater proportion of all presentations are detected for some EDs than others in this dataset. This would make admission rates for EDs with a high proportion of attendances detected look lower. There is no way to test this with the data available, however, the increase in the proportion of very short admissions seen in hospitals with higher admission rates supports the conclusion that these findings illuminate real differences in hospital admission practices.

While this study shows how admissions data differ to attendance data, it needs to be borne in mind that the majority of self-harm in the community, particularly self-injury, does not present to hospital services at all.[25] Hospital presenting self-harm is of interest as it is associated with high levels of psychiatric morbidity[26] as well as the physical harm caused and has been shown to be an important risk factor for future suicide.[4] However, the processes that determine whether someone who has self-harmed in the community presents to the ED may

well vary between population groups and geographical areas, meaning that even ED attendance data may not fully represent true variations in rates of self-harm in the community between areas.

## Implications

Differences in likelihood of admission between hospitals will bias estimates of rates of self-harm for different areas. This has the potential to exacerbate health inequalities if it results in resources being directed away from disadvantaged areas and populations on the basis of an underestimate of rates. Previous research has found lower rates of admission for self-harm in English city centres compared with the suburbs.[6 16] This study suggests such findings may be partly explained by admission practices resulting in an underestimate of self-harm in the inner cities when admission data are used. Underestimation of self-harm rates in this study mainly affected the boroughs of Lambeth and Southwark. These boroughs include areas with high levels of deprivation and substantial black and minority ethnic populations and experience higher rates of adverse health outcomes[27] including higher rates of lifetime suicidal behaviours than the national average on community surveys.[28] Current reliance of public health services in these boroughs on self-harm admissions data to formulate their suicide and self-harm prevention strategies[29] risks failing to identify need in already disadvantaged populations.

If the hospital practices driving lower admissions are more typical of hospitals serving more deprived inner-city areas or otherwise under greater resource pressure, it is likely that these patterns are being replicated elsewhere in the country. At a national level, London has much lower rates of self-harm admissions than the English average,[15] however, these findings provide reason to be cautious in interpreting this as meaning there is truly lower underlying risk in the capital. Likewise, in other settings, research findings and public health planning based on admissions data need to be alert to the potential influence of such biases.

Routinely collected data on attendances to medical services following self-harm will always have an important role in research and public health planning both in England and internationally. They provide more comprehensive coverage and regular updates than research datasets can, allowing variations between areas and over time to be examined. Such data also have the potential to increase clinical services' understanding of the populations they serve and help them configure services more appropriately. This study suggests that routine data covering ED attendances would be more appropriately used for these purposes than admissions data. Such a dataset was included as a 'placeholder indicator' in Public Health England's PHOF from 2015[11] as a statement of intent to begin using such data as soon as it became available. Unfortunately, the lack of a reliable source for the measure means it has never been produced and is now earmarked for removal in the next iteration of the PHOF.[30] The findings of this paper suggest that efforts to find a way to create a reliable ED self-harm dataset should remain a priority. The widespread use of electronic health records by mental health trusts and their increasing incorporation into linked research databases through systems including CRIS may provide avenues to do this in future, especially if linkage can be extended to ED clinical records.

## CONCLUSIONS

Currently in England, as in many other countries, hospital admissions are the only comprehensive, reliably coded data on the incidence of non-fatal self-harm available and so are widely used in research and as a public health indicator. This analysis demonstrates that doing so may risk underestimating relative rates in inner-city areas and so exacerbating existing health inequalities. Hospitals differ substantially in the proportions of individuals attending EDs with self-harm who get admitted. These differences are not explained by patient characteristics, type of self-harm or indicators of the severity of self-harm which suggests differences in hospital policies and practices are key. ED attendances for self-harm would provide a less biased estimate of area rates for comparisons hence making such data routinely available should be a public health priority.

**Acknowledgements** We are grateful for the advice provided regarding local hospital practices by Sean Cross, consultant liaison psychiatrist in SLaM and Clinical Director of the Mind and Body Programme in SLaM & King's Health Partners Academic Health Sciences Centre and Gopinath Ranjith, consultant liaison psychiatrist in SLaM.

**Contributors** CP devised the initial research question, carried out statistical analyses, drafted the manuscript and revised it in response to reviewer comments. IB advised on and supervised the statistical analyses and read and commented on the manuscript. MH and SH helped refine the initial research question, contributed to the interpretation of results and read and commented on the manuscript.

**Funding** CP is funded by a Welcome Trust Research Training Fellowship (grant number 105757/Z/14/Z). This paper represents independent research part funded by the National Institute for Health Research (NIHR) Biomedical Research Centre at South London and Maudsley NHS Foundation Trust and King's College London.

**Disclaimer** The views expressed are those of the authors and not necessarily those of the NHS, the NIHR or the Department of Health and Social Care.

**Map disclaimer** The depiction of boundaries on the map(s) in this article does not imply the expression of any opinion whatsoever on the part of BMJ (or any member of its group) concerning the legal status of any country, territory, jurisdiction or area or of its authorities. The map(s) are provided without any warranty of any kind, either express or implied.

**Competing interests** None declared.

**Patient consent for publication** Not required.

**Ethics approval** Research using data within CRIS is covered by a database approval from Oxford REC C (18/SC/0372), this project was approved by the CRIS Oversight Committee (Project number 14-026). Linkage of HES data to CRIS was given approval by the NHS Health Research Authority Confidentiality Advisory Group under Section 251 of the NHS Act 2006 in 2011 (reference ECC 3-04(f)/2011).

**Provenance and peer review** Not commissioned; externally peer reviewed.

**Data availability statement** Data may be obtained from a third party and are not publicly available.

**ORCID iD**
C Polling http://orcid.org/0000-0003-2657-0696

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
