## [Reviewer comments · BMJ Open]

ARTICLE DETAILS

TITLE (PROVISIONAL)	Differences in hospital admissions practices following self-harm and their influence on population-level comparisons of self-harm rates in South London: an observational study.
AUTHORS	Polling, C; Bakolis, Ioannis; Hotopf, Matthew; HATCH, STEPHANI

VERSION 1 – REVIEW

REVIEWER	Chris Metcalfe University of Bristol United Kingdom
REVIEW RETURNED	31-Jul-2019

GENERAL COMMENTS	In this manuscript the authors compare emergency department attendances and hospital admissions as measures of the incidence of self-harm, and highlight that due to variation in the proportion of attendees who are admitted, comparing geographic areas using HES (admission) data could be misleading. This research confirms that the variation between hospitals in the proportion of self-harm presentations who are admitted is also observed in London (for previous studies see e.g. R Carroll, C Metcalfe, D Gunnell. Hospital management of self-harm patients and risk of repetition: Systematic review and meta-analysis. Journal of Affective Disorders 2014; 168:476-483). I hope the authors find the following specific comments to be useful: [1] GENERAL The manuscript cautions against using HES admissions data to guide policies for the management of self-harm. Are there any examples the authors can give of this happening? At least amongst researchers, this weakness of admissions data as a measure of self-harm incidence is very well known. [2] ABSTRACT - CONCLUSIONS & MAIN TEXT CONCLUSIONS A comparison of four hospitals does not strongly support conclusions about the particular characteristics of hospitals in inner-city areas. MINOR Page 7, line 32. Should be "Data were" rather than "Data is". Page 9, line 50. Should it be "(SRRs)" rather than "(SARs)"? Page 10, line 37. "living in the study area were identified" is repeated. Table 1. There is a missing percentage for total males in the admissions column. Supplementary Figure 1. The y-axis is labelled as "SAR" rather than "SRR".
--

REVIEWER	keith waters Honorary Research Fellow (Self-Harm/Suicide Prevention) & Director of Centre for Self Harm and Suicide prevention, Centre for Research and Development, Derbyshire Healthcare NHS Foundation Trust Kingsway Hospital, Kingsway, Derby, DE22 3LZ UK T DoH funded MCM study team member
REVIEW RETURNED	25-Aug-2019

GENERAL COMMENTS	This is a comprehensive and detailed piece of work which adds to our understanding of how to obtain reliable data related to self-harm hospital presentations. It clearly meets its objectives and adds to the existing body of knowledge in this area. Whilst the findings are important in the public health arena and regarding service planning, as a clinician I would have liked a little recognition that relates to the importance in terms of clinical delivery and suicide prevention approaches. I guess what I mean by this is related to “why is it important that we obtain accurate local data on self-harm presentations?” this in order to help mould self-harm services as well as work in suicide prevention.  • Firstly if we are able to obtain detailed clinical personal and demographic information related to all hospital presentations this will help to increase our knowledge and awareness of this complex clinical presentation . • Secondly, we can try to ensure that services and approaches are geared to meet the diverse needs of this population group. The attendance at hospital not only is important in ensuring the physical and medical needs of the individual are addressed but also the attendance is the opportunity for a full and detailed biopsychosocial assessment to be able to explore the background factors needs and problems that the individual is facing, which in turn can help to mould which further help and input and suicide prevention approaches could be of assistance. Although we still need to know more about self-harm assessment, the current body of knowledge does highlight the importance of having a self-harm assessment. This supported by nice guidelines. Therefore, having reliable and accurate real time data on all self-harm presentations to hospital is important clinically as well as in understanding local profiles. As the authors say, currently in England, as in many other countries, hospital admissions are the only comprehensive, reliably coded data on the incidence of non-fatal self-harm available and so are widely used in research and as a public health indicator. Having been privileged to be part of the MCM team, as noted in this paper, there are problems relying on this approach to estimate Self Harm local rates This study demonstrates that doing so risks underestimating relative rates and has the potential to exacerbate existing health inequalities. It is further interesting that the Hospitals differed substantially in the proportions of individuals attending EDs with self-harm who get admitted and that these differences are not explained by patient characteristics, type of self-harm or indicators of the severity of self-harm. This has led the authors backed by the discussion with liaison team staff to suggests differences in hospital policies and practices are key. This paper and the works from the MCM study and that of Ireland would support the stance that ED attendances for self-harm would provide a less biased estimate of area rates for comparisons hence making such data routinely available should be a public health priority.
--

	Routinely collected data on attendances to medical services following self-harm will always have an important role in research and public health planning both in England and internationally. This study suggests that routine data covering ED attendances would be more appropriately used for these purposes Whilst the findings of this paper suggest that efforts to find a way to create a reliable ED self-harm data-set should remain a priority. They suggest that the widespread use of electronic health records by mental health trusts and their increasing incorporation into linked research databases through systems including CRIS may provide avenues to do this in future. This approach in order to capture all attendances to ED for self-harm does require a robust system ensuring a referral to the mental health services for every attendance. Whilst in the paper a description of the use of the “audit” showed benefits of the CRIS data set, it does note that not all attendances result in referrals to mental health services , perhaps the way forward should be working towards that unified recording of self-harm a tendencies to ED and combining that information and data sets with those of mental health services and the admission data sets . In order to understand the differences across the four hospitals in the admission rates the use of the liaison teams has provided a useful insight. This may however present a slightly skewed perspective i.e. that of the mental health services. Whilst this is an appropriate route to try to understand these differences, I feel the understanding could have been strengthened by obtaining the perspectives from the 4 hospitals including their Ed departments. I hope my observations are of help to the authors, they are observations and not indications of any major changes needed to the paper as I feel it has enough strengths as it is to be published. I would like to thank the authors for the time and effort they have put into this which will in the long term help our understanding of this complex area and help to influence future service deliveries and suicide prevention approaches.
--	---

REVIEWER	Timothy Schmutte Yale School of Medicine, Department of Psychiatry USA
REVIEW RETURNED	30-Aug-2019
GENERAL COMMENTS	This study provides a unique contribution to the body of knowledge on variations in hospital management of self-harm.

VERSION 1 – AUTHOR RESPONSE

Reviewer: 1

Thank you for the useful comments that have helped me improve this manuscript. Please see how I have made changes related to your specific concerns below.

[1] GENERAL

The manuscript cautions against using HES admissions data to guide policies for the management of self-harm. Are there any examples the authors can give of this happening? At least amongst researchers, this weakness of admissions data as a measure of self-harm incidence is very well known.

The manuscript focuses on the use of self-harm data for monitoring rates, comparing areas and allocating resources at the population level and does not make assertions about the management of self-harm in individuals. I describe in the background and discussion the widespread use of self-harm admissions data within public health in England including its use as the self-harm indicator in the Public Health Outcomes Framework (ref 11) and in area profiles (ref 12) and reference an example from one of the boroughs in the study area (Lambeth) of how this results in local public health teams using this data to understand need locally (ref 29).

I agree that researchers have been more cautious about using admissions data to represent self-harm in the population. However, the only academic paper reporting England-wide comparison of rates of self-harm between areas I am aware of (ref 5) has relied on this data as the only source available. While I have not cited them to avoid confusing the focus of the manuscript, there is also a larger body of research addressing other questions that uses linkage to clinical records to identify self-harm as an outcome, which again relies on admission as the only reliably coded data. (Examples include 1. Mars B, Cornish R, Heron J, et al. Using data linkage to investigate inconsistent reporting of self-harm and questionnaire non-response. Archives of suicide research 2016;20(2):113-41. 2. Singhal A, Ross J, Seminog O, et al. Risk of self-harm and suicide in people with specific psychiatric and physical disorders: comparisons between disorders using English national record linkage. Journal of the Royal Society of Medicine 2014;107(5):194-204.).

[2] ABSTRACT - CONCLUSIONS & MAIN TEXT CONCLUSIONS

A comparison of four hospitals does not strongly support conclusions about the particular characteristics of hospitals in inner-city areas.

I agree that we need to be cautious interpreting data that comes from four hospitals. I have amended in language in the abstract conclusions to:

“Public health policy that directs resources based on self-harm admissions data could exacerbate existing health inequalities in inner-city areas where this data may underestimate rates relative to other areas.”

I have amended the language in the main text conclusions to:

“This analysis demonstrates that doing so may risk underestimating relative rates in inner city areas and so exacerbating existing health inequalities.”

MINOR

Page 7, line 32. Should be "Data were" rather than "Data is".

I have made this amendment.

Page 9, line 50. Should it be "(SRRs)" rather than "(SARs)"?

I have made this amendment.

Page 10, line 37. "living in the study area were identified" is repeated.

The repetition has been deleted.

Table 1. There is a missing percentage for total males in the admissions column.

This has been added.

Supplementary Figure 1. The y-axis is labelled as "SAR" rather than "SRR".

This has been altered

Reviewer: 2

Thank you for your thoughtful comments on the manuscript and the work it describes. I am grateful for your assessment that the paper could be published as it is and does not require major changes in response to them. As such, I have not provided a point by point response, but have amended the implications section of the discussion to reflect some of the points you raise. In particular I include reference to the clinical as well as public health usefulness of reliable routine data on self-harm and your important point that the inclusion of ED clinical data in linked research datasets like CRIS would enhance their use in creating routine datasets of ED attendances for self-harm.

The final paragraph of this section now reads:

"Routinely collected data on attendances to medical services following self-harm will always have an

important role in research and public health planning both in England and internationally. They provide more comprehensive coverage and regular updates than research datasets can, allowing variations between areas and over time to be examined. Such data also have the potential to increase clinical services' understanding of the populations they serve and help them configure services more appropriately. This study suggests that routine data covering ED attendances would be more appropriately used for these purposes than admissions data. Such a dataset was included as a "placeholder indicator" in Public Health England's Public Health Outcomes Framework (PHOF) from 2015/16 as a statement of intent to begin using such data as soon as it became available. Unfortunately, the lack of a reliable source for the measure means it has never been produced and is now earmarked for removal in the next iteration of the PHOF30. The findings of this paper suggest that efforts to find a way to create a reliable ED self-harm dataset should remain a priority. The widespread use of electronic health records by mental health trusts and their increasing incorporation into linked research databases through systems including CRIS may provide avenues to do this in future, especially if linkage can be extended to ED clinical records."

Reviewer: 3

Thank you for your supportive comments, there were no required changes indicated in them.

VERSION 2 – REVIEW

REVIEWER	Chris Metcalfe University of Bristol, UK.
REVIEW RETURNED	09-Sep-2019
GENERAL COMMENTS	The authors have addressed all of my comments on the previous version.